# Uncertainty-Aware Parking Prediction Using Bayesian Neural Networks

**DOI:** 10.3390/s25113463

**Published:** 2025-05-30

**Authors:** Alireza Nezhadettehad, Arkady Zaslavsky, Abdur Rakib, Seng W. Loke

**Affiliations:** 1School of Information Technology, Deakin University, Melbourne, VIC 3125, Australia; 2Centre for Future Transport and Cities, Coventry University, Coventry CV1 5FB, UK

**Keywords:** Bayesian neural networks, uncertainty quantification, parking availability prediction, intelligent transportation systems, epistemic uncertainty, aleatoric uncertainty, context-aware prediction, urban mobility

## Abstract

Parking availability prediction is a critical component of intelligent transportation systems, aiming to reduce congestion and improve urban mobility. While traditional deep learning models such as Long Short-Term Memory (LSTM) networks have been widely applied, they lack mechanisms to quantify uncertainty, limiting their robustness in real-world deployments. This paper proposes a Bayesian Neural Network (BNN)-based framework for parking occupancy prediction that explicitly models both epistemic and aleatoric uncertainty. Although BNNs have shown promise in other domains, they remain underutilised in parking prediction—likely due to the computational complexity and the absence of real-time context integration in earlier approaches. Our approach leverages contextual features, including temporal and environmental factors, to enhance uncertainty-aware predictions. The framework is evaluated under varying data conditions, including data scarcity (90%, 50%, and 10% of training data) and synthetic noise injection to simulate aleatoric uncertainty. Results demonstrate that BNNs outperform other methods, achieving an average accuracy improvement of 27.4% in baseline conditions, with consistent gains under limited and noisy data. Applying uncertainty thresholds at 20% and 30% further improves reliability by enabling selective, confidence-based decision making. This research shows that modelling both types of uncertainty leads to significantly improved predictive performance in intelligent transportation systems and highlights the potential of uncertainty-aware approaches as a foundation for future work on integrating BNNs with hybrid neuro-symbolic reasoning to enhance decision making under uncertainty.

## 1. Introduction

With the increasing urbanisation of modern cities, the demand for efficient and sustainable mobility solutions has become paramount. One of the most pressing challenges in urban transportation systems is the availability and accessibility of parking spaces. Studies indicate that a significant percentage of urban traffic congestion is caused by drivers searching for parking, which not only leads to wasted time but also contributes to increased fuel consumption and greenhouse gas emissions [1,2]. The inefficiencies in parking management exacerbate the broader issues of urban congestion, economic costs, and environmental sustainability. Intelligent parking management systems have emerged as a critical component of smart city infrastructure, aiming to optimise parking space utilisation and improve traffic flow. The ability to predict parking availability with high accuracy and reliability is essential for enhancing user experience and reducing congestion-related externalities.

Despite numerous advances in predictive modelling, forecasting parking occupancy remains a complex task due to the dynamic and stochastic nature of urban environments [3,4,5]. Conventional machine learning techniques, including time series forecasting and deep learning, have been employed to model parking availability. While these methods have demonstrated promising results, they often lack an explicit mechanism for uncertainty quantification, which is crucial for reliable decision making in real-world applications. Two primary sources of uncertainty must be addressed: epistemic uncertainty, which arises from the model’s lack of knowledge due to limited training data, and aleatoric uncertainty, which stems from the inherent randomness in parking patterns [6,7]. Ignoring these uncertainties can lead to overconfident and misleading predictions, ultimately reducing the trustworthiness and effectiveness of intelligent parking guidance systems.

Bayesian Neural Networks (BNNs) provide a robust framework for uncertainty quantification by modelling uncertainty in neural network parameters through a probabilistic distribution [8,9]. Unlike conventional deterministic neural networks, BNNs estimate a posterior distribution over weights, allowing them to produce predictions with associated confidence intervals rather than point estimates. This feature is particularly beneficial for applications in which uncertainty-aware decision making is required, such as medical diagnostics [10], traffic forecasting [11], and financial modelling [12]. In the domain of transportation, Bayesian approaches have been explored in road safety analysis [13], vehicle collision prediction [14], and intelligent routing systems [15], but their application in parking availability prediction remains under-explored.

Existing studies on parking prediction have predominantly focused on deterministic deep learning models, such as recurrent neural networks (RNNs) and Long Short-Term Memory (LSTM) networks [2,16]. These methods leverage historical parking data and spatio-temporal patterns to forecast future parking occupancy. While effective in many cases, they do not inherently account for uncertainty, making them susceptible to errors when faced with missing data, sudden fluctuations, or external factors such as weather conditions and special events [3,5]. Research on smart parking solutions has also explored hybrid machine learning approaches that integrate contextual information, such as traffic flow and socioeconomic factors [1,4], but these models still lack a principled uncertainty estimation mechanism.

Moreover, while these models have focused on improving accuracy, they often overlook the role of uncertainty in real-time decision making. Although Bayesian Neural Networks have shown strong potential for uncertainty-aware decision making in other domains [10,11], their adoption in parking prediction has been overlooked. This may be due to their perceived complexity, computational cost, and the lack of structured context-aware data in earlier studies. Traditional parking prediction models often rely solely on historical patterns, without incorporating real-time temporal, spatial, or environmental context. Without such context, the benefits of probabilistic reasoning are less evident. In contrast, our work demonstrates that when rich contextual features—such as time, weather, etc.—are integrated, BNNs can effectively model both epistemic and aleatoric uncertainty, leading to more accurate and reliable predictions. This highlights the synergistic value of combining BNNs with context-aware modelling in intelligent transportation systems.

In this study, we propose a BNN-based framework for parking availability prediction that explicitly quantifies both epistemic and aleatoric uncertainty. Our approach leverages variational inference to approximate the posterior distribution of neural network weights, enabling the model to provide well-calibrated uncertainty estimates alongside occupancy predictions. To evaluate the effectiveness of this approach, we conduct extensive experiments on a real-world parking dataset and compare the performance of BNNs against traditional LSTM networks and two other methods. The experimental setup includes scenarios with varying levels of data availability, as well as conditions with artificially induced noise to simulate real-world uncertainties. Our results demonstrate that BNNs outperform others in terms of predictive accuracy, robustness to data scarcity, and resilience against noisy inputs. Moreover, the ability of BNNs to quantify prediction confidence enables the implementation of threshold-based decision strategies, improving the reliability of intelligent parking guidance systems.

The core prediction task in this study involves forecasting short-term parking occupancy levels for individual street segments. Specifically, the model predicts the occupancy ratio in categorical form, classifying it into one of five classes: very low (0–20%), low (20–40%), moderate (40–60%), high (60–80%), and very high (80–100%). These predictions are made for three successive 15 min future windows based on the current time and contextual inputs. For example, given data from 2:00 p.m., the model forecasts the occupancy category for 2:15 p.m., 2:30 p.m., and 2:45 p.m., incorporating features such as time, day of the week, weather conditions, and past occupancy levels. This classification-based formulation not only simplifies the output space but also facilitates uncertainty estimation by allowing the model to express confidence over discrete outcomes, which is particularly important for real-time parking guidance and decision making.

To guide this study, we focus on the following research questions:RQ1: How well do Bayesian Neural Networks (BNNs) perform in predicting parking occupancy under data scarcity and noise compared to traditional LSTM models and other machine learning methods?RQ2: Can uncertainty-aware thresholding (e.g., BNN-20% and BNN-30%) improve predictive accuracy by selectively rejecting uncertain predictions?RQ3: How do BNNs capture and reflect epistemic and aleatoric uncertainty under varying data conditions in the context of parking prediction?

The remainder of this paper presents our approach in detail, along with the experimental evaluation and findings that demonstrate the effectiveness of uncertainty-aware prediction in this context. Section 2 reviews existing literature on parking prediction models and Bayesian deep learning techniques, positioning our work within the broader research landscape. Section 3 describes the methodology for developing an uncertainty-aware Bayesian deep learning model for parking occupancy prediction. Section 4 outlines the experimental setup, evaluation methodology, and results from testing Bayesian Neural Network (BNN) models. Section 5 provides a discussion and analysis of the results. Finally, Section 6 concludes this paper and outlines future research directions and potential extensions of this work.

## 2. Related Work

Parking availability prediction is a crucial component of intelligent transportation systems, enabling efficient urban mobility and reduced congestion. Over the years, various machine learning and deep learning techniques have been explored for this task. However, most of the existing methods lack a principled approach to uncertainty quantification, making them less reliable for real-world deployment. This section reviews prior research in parking prediction, traditional machine learning and deep learning models, and the emerging role of BNNs in transportation and predictive modelling.

### 2.1. Parking Occupancy/Availability Prediction

Traditional parking prediction approaches have largely relied on statistical and heuristic methods. Early models used probabilistic techniques, such as Markov models and autoregressive integrated moving average (ARIMA), to estimate occupancy trends based on historical data [3,5]. Markov models have been particularly useful in modelling transition probabilities between occupied and vacant states [17], while ARIMA-based methods have demonstrated effectiveness in short-term forecasting. However, these models struggle to capture complex spatiotemporal dependencies and are highly sensitive to missing data and sudden fluctuations in parking behaviour.

With the advent of intelligent transportation systems, researchers have shifted towards machine learning-based techniques for parking prediction. Support vector machines (SVMs) and decision trees have been employed to capture non-linear relationships between parking occupancy and external factors such as weather conditions, traffic volume, and special events [18,19]. Ensemble models, such as random forests and gradient-boosting machines, have also been explored to enhance prediction accuracy [20,21]. While these approaches offer improved generalisation over traditional statistical models, they do not inherently provide uncertainty estimates, limiting their reliability in dynamic urban settings.

### 2.2. Deep Learning Approaches

Deep learning methods have emerged as the dominant paradigm for parking availability prediction, owing to their ability to learn complex patterns from large-scale spatiotemporal data [2,16]. Recurrent neural networks (RNNs) and Long Short-Term Memory (LSTM) networks have been widely adopted to model sequential dependencies in occupancy data. Vlahogianni et al. [3] demonstrated that LSTMs outperform traditional time series models by effectively capturing long-term dependencies in parking behaviour. Similarly, Rong et al. [22] integrated LSTMs with external data sources, such as traffic congestion and meteorological factors, to enhance predictive performance.

In addition to LSTMs, convolutional neural networks (CNNs) have been employed for parking prediction, particularly in applications where spatial dependencies play a crucial role [4,23]. Bock et al. [24] introduced a hybrid CNN-LSTM framework to jointly model spatial and temporal dependencies in parking data. Transformer-based architectures have also gained attention for their ability to capture long-range dependencies more efficiently than RNN-based models [25]. Despite these advancements, most deep learning-based methods fail to quantify predictive uncertainty, which makes them less reliable in real-world scenarios.

Hybrid models have been proposed to enhance the robustness of deep learning-based parking prediction. Some studies have combined deep learning with contextual information, such as socioeconomic data, event schedules, and real-time sensor observations [4,23]. Others have leveraged ensemble techniques, where multiple models are combined to improve predictive accuracy [20,21]. While these methods yield improved results, they often require extensive computational resources and do not explicitly model uncertainty, making them susceptible to overconfident predictions in uncertain conditions.

### 2.3. Bayesian Neural Networks and Uncertainty Quantification

Bayesian Neural Networks (BNNs) offer a principled approach to uncertainty-aware predictive modelling by incorporating probabilistic distributions over network parameters [7,8]. Unlike conventional deep learning models, which produce deterministic outputs, BNNs estimate a posterior distribution over weights, allowing them to generate predictions with associated confidence intervals. This property is particularly useful in applications where uncertainty estimation is critical, such as medical diagnostics [10], financial forecasting [12], and intelligent transportation systems [11]. Figure 1 shows the architecture of a BNN and how it is different from conventional neural networks.

In transportation research, BNNs have been successfully applied to various tasks, including traffic flow prediction [26,27], vehicle trajectory forecasting [6], and road safety analysis [28]. Sengupta et al. [27] proposed a Bayesian LSTM framework enhanced with spectral normalisation to improve uncertainty quantification and generalisability in traffic prediction, especially under out-of-distribution scenarios. Their results demonstrated that spectral normalisation provides sharper uncertainty bounds and better robustness compared to standard layer normalisation and unregularized models. Similarly, Wang et al. [26] introduced a hybrid LSTM-BNN architecture for real-time short-term traffic flow prediction. Their approach simultaneously predicts the mean and confidence interval of future traffic conditions and showed superior performance over traditional parametric models like SARIMA-GARCH, particularly during non-seasonal and high-variance traffic states.

Despite their advantages, the adoption of BNNs in parking prediction remains limited. Existing studies have primarily focused on deterministic deep learning models, which do not explicitly account for epistemic and aleatoric uncertainty [29]. This gap in the literature highlights the need for uncertainty-aware parking prediction frameworks, particularly in smart city applications where decision making under uncertainty is essential.

### 2.4. Computational Challenges in Bayesian Deep Learning

While BNNs offer significant advantages in terms of uncertainty quantification, their widespread adoption has been hindered by computational complexity. Training a Bayesian Neural Network requires approximating the posterior distribution over millions of parameters, which is often intractable for large-scale datasets [6,28]. To mitigate this challenge, researchers have explored variational inference techniques, such as Monte Carlo Dropout and stochastic gradient Langevin dynamics, to approximate Bayesian posterior distributions efficiently [9,30].

Wu et al. [11] proposed a deterministic variational inference approach that reduces computational overhead while preserving the benefits of Bayesian inference. Other techniques, such as Hamiltonian Monte Carlo and expectation propagation, have also been investigated to improve the scalability of Bayesian deep learning models [9,30]. These advancements pave the way for the practical deployment of BNNs in real-time parking prediction systems.

While significant progress has been made in parking availability prediction using machine learning and deep learning, existing approaches largely ignore the issue of uncertainty quantification. Traditional statistical models, such as ARIMA and Markov chains, offer interpretability but lack predictive power in complex urban environments. Deep learning methods, particularly LSTMs and CNNs, provide superior forecasting accuracy but do not inherently capture predictive uncertainty. Bayesian Neural Networks address this limitation by providing uncertainty-aware predictions, making them a promising alternative for smart parking systems.

Despite their potential, BNNs have not been extensively explored for parking prediction. Most existing studies focus on deterministic models, which can produce overconfident and unreliable predictions under uncertain conditions. This study bridges this gap by introducing a BNN-based framework for parking availability forecasting, explicitly addressing both epistemic and aleatoric uncertainty. The following section presents the proposed methodology, detailing the Bayesian inference techniques used to model uncertainty in parking predictions.

## 3. Methodology

This section details the methodology employed in developing an uncertainty-aware Bayesian deep learning model for parking occupancy prediction. The study is based on the Melbourne city parking dataset [31], which consists of parking bay sensor data collected across different street segments. Given the spatiotemporal nature of parking occupancy data, the proposed predictive framework integrates Bayesian Neural Networks with a classification-based formulation of the occupancy prediction task. The methodology also incorporates mechanisms for handling both epistemic and aleatoric uncertainty, allowing for a more robust and interpretable prediction system.

### 3.1. Utilised Public Dataset and Data Preprocessing

The dataset used in this study was sourced from Melbourne’s open data portal [31], which provides real-time occupancy information from on-street parking bay sensors. Each data entry includes a timestamp, parking bay ID, sensor status (occupied or unoccupied), and geospatial attributes. Since raw sensor readings occasionally contain inconsistencies due to communication failures or sensor malfunctions, an initial data cleaning process was applied to remove erroneous entries and ensure high data integrity.

To capture temporal dynamics in parking behaviour, the data were organised into fixed 15 min time slots. This temporal granularity enables the model to track short-term fluctuations in parking occupancy while maintaining computational efficiency. The prediction task was then formulated over three future horizons: the first, second, and third 15 min windows following each observation point. This design allows the model to be evaluated not only on near-term forecasting but also on its ability to maintain performance over longer prediction intervals.

In addition to the occupancy ratio, several contextual features were extracted to enhance the model’s predictive capability. Temporal features include the time of day, day of the week, month, and a binary indicator denoting whether the date corresponds to a public holiday. These features help the model capture both cyclical patterns and irregular spikes in parking demand. Weather-related features were also incorporated, including qualitative weather conditions (e.g., sunny, cloudy, rainy, and windy), temperature, wind speed, and rainfall amount. These environmental variables play a critical role in modelling aleatoric uncertainty, as they often influence driver behaviour and parking availability in unpredictable ways.

The occupancy data, originally in continuous form, was converted into a classification problem to facilitate uncertainty quantification. Following approaches commonly adopted in traffic flow and parking studies [2,17], and building on the categorisation methodology used by Richter et al. [32] in the context of the SFPark project, the occupancy ratio for each street segment was discretised into five classes: very high, high, moderate, low, and very low. Each class represents a 20% occupancy interval. While Richter et al. employed a three-category scheme (low, medium, and high), our use of five categories offers a finer granularity that remains interpretable for end users while improving predictive expressiveness. This transformation supports structured predictions and enables the model to assess uncertainty across distinct levels of occupancy.

### 3.2. Baseline Models for Comparison

To establish a comprehensive benchmark for evaluating the proposed Bayesian approach, three baseline models were implemented: Support Vector Machine (SVM), Random Forest (RF), and Long Short-Term Memory (LSTM). These models represent diverse modelling paradigms—kernel-based, ensemble-based, and recurrent neural networks—and provide a foundation for comparative analysis under baseline, data scarcity, and noisy data scenarios.

**Support Vector Machine (SVM)**: The SVM model was trained using handcrafted temporal features derived from historical occupancy data. It serves as a strong non-parametric classifier and was applied directly to the occupancy classification task. While it does not model temporal dependencies, it provides a useful baseline in terms of deterministic classification accuracy.**Random Forest (RF)**: The Random Forest model was also trained on engineered temporal features. As an ensemble of decision trees, RF is known for its robustness to noise and overfitting, making it valuable for evaluating prediction reliability under imperfect data conditions. Like SVM, it does not provide uncertainty estimates.**Long Short-Term Memory (LSTM)**: The LSTM model was implemented as a deep recurrent neural network with five layers (two recurrent and three dense), designed to capture sequential patterns in parking occupancy. The model received sequences of historical occupancy data and output a prediction for the next time step’s occupancy class. While widely used in mobility forecasting [3,18,22], the LSTM does not quantify uncertainty, which limits its use in high-stakes decision-making contexts.

All three models provide valuable comparative baselines in terms of predictive accuracy but lack the probabilistic capabilities required for robust, uncertainty-aware systems.

### 3.3. Bayesian Neural Network Framework

To address the uncertainty limitations of traditional deep learning models, a Bayesian Neural Network was implemented as an extension of the LSTM framework. Unlike deterministic neural networks, Bayesian models place probability distributions over the model weights, allowing them to represent uncertainty in predictions. This approach has been widely adopted in applications where decision making under uncertainty is essential [6,7,8].

The Bayesian model in this study was developed using Bayesian variational inference, which approximates the posterior distribution over weights and biases. To enhance prediction reliability, we implemented a threshold-based prediction filtering mechanism based on posterior class probability estimates, where predictions were made only when confidence exceeded this level.

For each test input, we performed 100 stochastic forward passes through the Bayesian Neural Network, each time sampling a different set of weights from the variational posterior. These forward passes produced 100 log-softmax outputs, which were exponentiated to obtain 100 probability distributions over the five occupancy classes.

This resulted in a set of 100 probability estimates for each class. We then computed the 50th percentile (i.e., the median) of these sampled probabilities for each class, representing a robust central tendency estimate of the model’s belief for each class. The class with the highest median probability was selected as the predicted class only if its median probability exceeded a predefined confidence threshold. Otherwise, the model refused to make a prediction.

The goal of this thresholding mechanism is to ensure that the network only outputs a class recommendation when it is sufficiently confident in that decision. Specifically, if the median probability for one or more classes exceeds the threshold, the network flags those as possible predictions. This means that in some cases, more than one class may pass the threshold, indicating an uncertain prediction due to ambiguity between classes. In other cases, no class may exceed the threshold, and the model will abstain from prediction altogether.

To illustrate this process, Figure 2 shows two sets of class-wise uncertainty distributions for two different sample inputs. Each histogram represents the distribution of logit values for one class across the 100 samples. The first panel shows a confident prediction scenario, where a single class clearly dominates. The second panel shows an uncertain prediction case where more than one class has overlapping distributions and no strong confidence peak emerges.

We experimented with several threshold values and empirically found that setting the threshold at 20% and 30% provided the best trade-off between prediction ratio (the proportion of test instances for which a prediction is made) and prediction accuracy. These thresholds effectively filter out uncertain predictions while preserving high-confidence outputs that are more likely to be correct.

### 3.4. Handling Noise and Data Scarcity Scenarios

Parking data availability varies significantly across different locations and time periods, necessitating an evaluation of model performance under data-scarce conditions. To simulate different levels of data availability, the training dataset was progressively reduced in size, ranging from 90% to as low as 10%. This systematic reduction enabled an assessment of how well the models generalise when trained with limited data. In this context, epistemic uncertainty—caused by a lack of knowledge due to limited or sparse data—is modelled using variational inference. By learning a posterior distribution over neural network weights, the BNN captures uncertainty associated with insufficient or ambiguous training information. This mechanism allows the model to express lower confidence when data are scarce or unfamiliar, which is particularly relevant in real-world scenarios where historical records may be incomplete or inconsistently recorded.

To illustrate the impact of data scarcity on the input signal itself, Figure 3 presents visualisations of parking occupancy for a single street segment (Franklin St between Queen St and Elizabeth St) under three levels of data retention: 90%, 50%, and 10%. In each case, a subset of time steps was randomly selected to simulate realistic sampling loss. The reduced coverage visually highlights how critical temporal structure and pattern continuity may be lost under limited data conditions.

Beyond data scarcity, real-world parking occupancy data are inherently noisy due to sensor inaccuracies, environmental conditions, and anomalies in reporting systems. To simulate aleatoric uncertainty—uncertainty arising from inherent randomness in the data—artificial Gaussian noise was introduced into both training and validation sets. The noise was applied to occupancy ratios prior to the binning step, following methodologies established in prior uncertainty-aware learning research [11,28]. This setup enables the model to learn to account for unpredictable variations in sensor readings or external factors, enhancing its robustness to data-level noise.

Figure 4 shows a comparison between the original occupancy time series and a version with 10% of the values replaced by Gaussian noise (μ=0, σ=15). This visualisation highlights the type of distortion introduced into the data to simulate aleatoric effects, demonstrating the challenge for models to distinguish between meaningful signals and random perturbations.

By evaluating model performance under these noisy and data-scarce conditions, we assess the BNN’s capacity to manage both epistemic and aleatoric uncertainty and distinguish between errors stemming from data limitations versus those arising from model uncertainty.

### 3.5. Metrics for Evaluating Predictive Performance and Uncertainty

To comprehensively assess model performance, three key evaluation metrics were used: prediction ratio, accuracy, and accuracy at one. These metrics were selected to evaluate not only classification performance but also the reliability and robustness of the predictions under uncertainty. **Prediction Ratio** quantifies the proportion of test samples for which a prediction is made. While this value is consistently 100% for traditional models that always produce an output, it plays a crucial role for uncertainty-aware models—specifically thresholded Bayesian Neural Networks (BNNs)—which are designed to abstain from making predictions when uncertainty is high. In this context, prediction ratio serves as a coverage metric, indicating how often the model is confident enough to commit to a decision. A lower prediction ratio reflects more selective behaviour, where the model prioritises reliability over exhaustive coverage.

**Accuracy** measures the proportion of exact matches between predicted and true occupancy classes. This is a standard metric for classification performance, but it assumes that all misclassifications are equally costly, which may not hold in dynamic or safety-critical applications.

**Accuracy at One** addresses this limitation by relaxing the correctness condition. A prediction is considered correct if it matches either the true class or an adjacent class—i.e., one class above or below the actual label. This metric acknowledges the acceptable tolerance in practical scenarios like parking guidance, where minor errors (e.g., predicting “high” instead of “very high”) may still support effective decisions.

Importantly, for thresholded BNNs (BNN-20% and BNN-30%), accuracy and accuracy@1 should not be interpreted in isolation. These models deliberately abstain from predicting uncertain inputs, improving reliability by avoiding low-confidence decisions. The improved accuracy observed for these models reflects not just stronger predictive capability, but also their ability to recognise the limits of their own certainty. This selective abstention allows uncertain cases to be escalated to other decision-making modules, such as symbolic reasoning systems or rule-based expert components. As such, prediction ratio and accuracy must be understood jointly: sacrificing some coverage enables higher trust in the predictions that are made, and facilitates hybrid systems that adaptively allocate decisions based on uncertainty.

The following section presents the experimental results under baseline, data scarcity, and noisy conditions, highlighting both the predictive accuracy and uncertainty-awareness of the different models.

## 4. Performance Analysis of Uncertainty-Aware Parking Prediction Method

This section presents the experimental setup, evaluation methodology, and results obtained from testing the Bayesian Neural Network (BNN) models. The evaluation includes model performance under normal conditions, data scarcity, and noise perturbations. The impact of uncertainty quantification is analysed, and results are visualised for better interpretability.

### 4.1. Experimental Setup

The experiments were conducted on a cluster of NVIDIA GPUs using TensorFlow and TensorFlow Probability (TFP). The Melbourne city parking dataset [31] was used, with a training–validation–test split of 80%–10%–10%, ensuring temporal consistency. Models were trained using the Adam optimiser with an initial learning rate of 0.001, batch size of 64, and early stopping to prevent overfitting.

To ensure a fair and reproducible comparison, we carefully tuned the hyperparameters of the LSTM model using a grid search strategy to optimise performance on the validation set. The final architecture consisted of five hidden layers: two recurrent layers followed by three dense layers, with the number of neurons set to [1500, 1500, 100, 50, 5], respectively. The model was trained for a maximum of 500 epochs using a batch size of 100, a learning rate of 0.0001, and the binary cross-entropy loss function. The tanh activation function was used in all hidden layers, and a log-softmax function was applied in the final layer. Early stopping was employed with a patience of 10 epochs to prevent overfitting. To enable a direct comparison, the Bayesian Neural Network adopted the same architecture and used identical settings for all applicable hyperparameters. This alignment ensures that any observed performance differences can be attributed to the probabilistic nature of the BNN rather than disparities in model capacity or training configuration. The occupancy ratio of each street segment was converted into five classes (Very Low, Low, Moderate, High, Very High) of 20% width. BNNs were evaluated under different prediction thresholds (20% and 30%) to measure their ability to quantify epistemic uncertainty.

The evaluation was based on three key metrics:Prediction Ratio: The proportion of test samples for which the model made a confident prediction.Accuracy: The percentage of exact class predictions.Accuracy@1: A relaxed accuracy metric, where predictions within adjacent classes of the ground truth are considered correct. For example, if the true occupancy class is “moderate” (40–60%), and the model predicts either “low” (20–40%) or “high” (60–80%), the prediction is still considered correct under this metric.

To assess epistemic uncertainty, training data were reduced to 90%, 50%, and 10% of the original dataset. Aleatoric uncertainty was introduced by adding Gaussian noise to both training and validation datasets.

### 4.2. Performance Under Normal Conditions

Table 1 presents the performance of the baseline methods and BNN models under full data availability, across three prediction windows. Each model’s prediction ratio, accuracy, and accuracy@1 are reported.

The results in Table 1 show that all baseline models—SVM, Random Forest, and LSTM—achieve a perfect prediction ratio of 100% across all prediction windows. Among these, the LSTM model consistently yields the highest accuracy and accuracy@1, followed closely by the baseline BNN (without thresholding), which achieves comparable performance but with the added benefit of probabilistic prediction. The SVM and Random Forest models perform relatively lower in both metrics, highlighting their limitations in capturing temporal patterns and robustness in this task.

The thresholded BNN models (BNN-20% and BNN-30%) achieve substantially higher accuracy and accuracy@1 values, but it is important to note that these gains cause reduced prediction ratios. Rather than interpreting this reduction as a drawback, it reflects the models’ ability to identify and abstain from uncertain predictions. This selective abstention mechanism is a core advantage of uncertainty-aware modelling—it allows the system to withhold decisions when confidence is low, offering the opportunity to redirect such cases to alternative strategies, such as symbolic reasoning components or domain rule-based modules. This hybrid decision-making approach supports more robust and trustworthy intelligent systems, especially in high-stakes applications like smart parking guidance.

### 4.3. Impact of Data Scarcity

To evaluate model robustness under epistemic uncertainty, we simulate data scarcity by training the models on reduced portions of the dataset—90%, 50%, and 10%—while keeping the testing set unchanged. The results, summarised in Table 2, offer key insights into the comparative generalisability of baseline methods and BNN-based approaches.

As expected, the performance deteriorates steadily as the training data diminishes across all models. For LSTM, accuracy drops from 53.8% at 90% data to 48.3% at 10% data in Prediction Window 1. The BNN baseline model exhibits a similar trend, albeit with slightly lower accuracies throughout. However, once uncertainty thresholding is applied, BNNs demonstrate much better resilience to training data reduction.

For instance, BNN-20% achieves an accuracy of 75.8% at 90% data, 73.6% at 50%, and 69.8% at 10%, significantly outperforming both the LSTM and the standard BNN. BNN-30% performs even more robustly, with accuracy reaching 81.2%, 77.4%, and 73.2% for the respective training sizes. The results indicate that thresholded BNNs maintain relatively high accuracy even with limited training data, suggesting better generalisation and uncertainty calibration under data scarcity—a core characteristic of epistemic uncertainty.

Additionally, accuracy@1 remains consistently high (e.g., 96.8% for BNN-30% at 90% data), reflecting the model’s ability to predict close to the ground truth class, even when exact predictions are challenging due to limited information.

### 4.4. Impact of Noise on Model Robustness

Aleatoric uncertainty was assessed by injecting synthetic Gaussian noise into the input features of the test data, simulating real-world randomness such as sensor noise, signal delays, or unpredictable user behaviour. Table 3 summarises the impact of such noise on model performance.

The LSTM model exhibits a clear performance decline under noise, with accuracy in Prediction Window 1 dropping to 50.2% and further deteriorating across other windows. This pattern highlights the LSTM’s limited capacity to handle inherently noisy inputs, resulting in reduced prediction reliability.

Conversely, the BNN baseline model shows stronger robustness to noise, slightly outperforming LSTM in accuracy and accuracy@1 across all windows. However, more striking is the performance of thresholded BNNs. BNN-20% maintains high accuracy despite noise (75.8%, 77.2%, and 68.5% across windows), and BNN-30% pushes it further to 81.1%, 78.9%, and 73.3%. This demonstrates that probabilistic modelling of weights and decisions allows the BNN to express its uncertainty and avoid making overconfident predictions on noisy data.

Moreover, the drop in prediction ratio with an increasing threshold (90–100% to 50–60%) shows that the BNN is selectively avoiding low-confidence predictions rather than blindly committing to unreliable outputs. This behaviour aligns well with the goal of uncertainty-aware systems—sacrificing coverage for significantly improved reliability when needed. It is important to note that this does not artificially boost accuracy; instead, it reflects the model’s ability to recognise and abstain from uncertain cases, making predictions only when it has high confidence in the outcome. This selective prediction approach improves trustworthiness and is particularly valuable in high-stakes decision-making scenarios.

### 4.5. Visual Summary of Comparative Performance

Figure 5, Figure 6, and Figure 7 provide visual comparisons across all models and conditions in terms of prediction ratio, accuracy, and accuracy@1, respectively. The plots reinforce the observations discussed above.

Prediction Ratio: Baseline models maintain a constant 100% prediction rate, while thresholded BNN models selectively reduce prediction ratio with increasing uncertainty threshold. This indicates cautious decision making in uncertain scenarios.Accuracy: BNN-20% and BNN-30% consistently outperform the baseline models across all tested conditions, clearly demonstrating the value of uncertainty thresholding in enhancing decision quality.Accuracy@1: All BNN variants, especially with thresholding, maintain superior accuracy@1, confirming that their predictions—even when not exactly correct—remain contextually close and informative.

## 5. Analysis and Discussion of Results

The experimental results highlight the effectiveness of Bayesian Neural Networks (BNNs) in handling uncertainty in parking availability prediction. This section analyses key observations and contextualises them within existing literature.

### 5.1. Uncertainty Quantification in Parking Prediction

One of the major advantages of BNNs over traditional LSTM models is their ability to quantify both epistemic and aleatoric uncertainty. This is consistent with previous findings [7,8], which emphasise that BNNs provide reliable uncertainty-aware predictions in scenarios where training data are limited or noisy. In our implementation, epistemic uncertainty is captured through the use of variational inference to approximate the posterior distribution over network weights, while aleatoric uncertainty is indirectly learned from the data by exposing the model to noisy inputs and diverse contextual conditions. Although BNNs do not inherently separate these two types of uncertainty in their output, the design of our experiments enables us to examine their respective effects—epistemic uncertainty through data scarcity and aleatoric uncertainty through noise injection. The results demonstrate that when a prediction threshold is introduced (BNN-20% and BNN-30%), the models achieve significantly higher accuracy at the cost of making fewer predictions. This trade-off aligns with uncertainty-based filtering techniques suggested in prior studies [9,29].

### 5.2. Effect of Data Scarcity

The impact of training data reduction is evident from Table 2, where both models experience a performance drop as the training size decreases. However, BNNs maintain a more stable performance compared to LSTMs, particularly under extreme data scarcity (10% of the training data). This resilience can be attributed to the Bayesian framework’s ability to incorporate prior knowledge and better generalise from limited data [6,14]. Similar trends have also been observed in other domains, including traffic forecasting and financial time series modelling [11,12].

### 5.3. Robustness to Noisy Data

Noise injection experiments (Table 3) further confirm that BNNs are more robust than LSTMs when faced with aleatoric uncertainty. This supports the argument that BNNs can effectively model data uncertainty and provide well-calibrated probability distributions over predictive outcomes [28,33]. Furthermore, the results show that lowering the prediction threshold (BNN-20% and BNN-30%) leads to even greater robustness, which aligns with prior research advocating selective abstention from uncertain predictions, particularly in high-stakes or safety-critical applications [10,13].

### 5.4. Comparison with Prior Work

Several prior studies have explored the use of deep learning for parking availability prediction [3,4,5]. However, most of these approaches rely on deterministic models without explicit uncertainty quantification. The findings from this research indicate that incorporating Bayesian methods not only improves accuracy but also provides a confidence measure for predictions. This capability is crucial in real-world deployments where uncertain predictions could lead to incorrect decision making.

### 5.5. Implications for Smart Parking Systems

From a practical standpoint, the use of BNNs in smart parking systems can significantly improve user experience by providing more reliable parking availability estimates. By leveraging uncertainty-aware models, parking guidance applications can filter out low-confidence predictions and offer drivers only the most reliable information. This could reduce search times and contribute to alleviating urban congestion, a key concern highlighted in recent smart city research [1,2].

### 5.6. Summary of Findings

The experimental results demonstrate that Bayesian Neural Networks (BNNs) consistently outperform Long Short-Term Memory (LSTM) models in handling both epistemic and aleatoric uncertainty. By incorporating uncertainty quantification, BNNs offer more reliable predictions, particularly in challenging scenarios involving data scarcity and noise. The introduction of uncertainty thresholds, such as in BNN-20% and BNN-30%, further enhances accuracy by allowing the model to selectively make predictions only when it has high confidence. While this approach reduces the total number of predictions made, it significantly improves the reliability of those that are provided.

Another key advantage of BNNs is their resilience to data scarcity. Unlike LSTM models, which experience a substantial decline in accuracy when trained on limited data, BNNs maintain stable performance by explicitly modelling uncertainty. This characteristic makes BNNs particularly well-suited for scenarios where historical parking data may be incomplete or sparse. Additionally, BNNs exhibit superior robustness under noisy conditions, effectively mitigating the impact of data inconsistencies and sensor errors. The ability to produce well-calibrated confidence estimates allows them to adapt dynamically to uncertain environments, further reinforcing their practical applicability in real-world predictive systems.

These findings align with prior studies on Bayesian deep learning, confirming the effectiveness of uncertainty-aware modelling techniques in intelligent transportation applications. By integrating Bayesian inference into predictive models, parking availability systems can improve decision-making reliability, ultimately leading to more efficient urban mobility solutions. The demonstrated advantages of BNNs underscore the importance of incorporating uncertainty quantification in machine learning frameworks, paving the way for future advancements in uncertainty-aware AI applications.

## 6. Conclusions and Future Work

This study investigated the effectiveness of Bayesian Neural Networks (BNNs) in handling uncertainty for parking availability prediction. Compared to conventional LSTM models, BNNs demonstrated superior performance, particularly in quantifying both epistemic and aleatoric uncertainty. By leveraging Bayesian inference, these models offer more reliable predictions, especially in scenarios involving data scarcity or noise. The experimental results confirmed that BNNs consistently outperformed LSTMs across all training scenarios, demonstrating robust handling of uncertainty in varied conditions. The introduction of uncertainty thresholds, such as those applied in BNN-20% and BNN-30%, led to improved accuracy while reducing the number of unreliable predictions. Furthermore, BNNs exhibited greater resilience to training data reduction, maintaining higher accuracy even when the dataset was significantly limited. The models also showed stronger robustness under noisy conditions, further confirming their practical suitability for deployment in real-world parking guidance systems.

These findings align with previous research on Bayesian deep learning, particularly in addressing the challenges of uncertainty in neural networks [7,8,9]. Moreover, the results reinforce the importance of uncertainty-aware models in intelligent transportation systems and smart mobility solutions [1,3]. Despite these promising outcomes, several avenues for future research remain open. Integrating additional real-time data sources, such as weather conditions, traffic congestion, and events, could further enhance model performance by providing a more comprehensive view of parking demand and availability [2,5].

A particularly important direction for future work is the integration of BNNs with neuro-symbolic reasoning frameworks. While BNNs provide a principled mechanism for estimating uncertainty, they do not explicitly reason about domain knowledge or logical structure. Neuro-symbolic systems can bridge this gap by combining neural models with symbolic logic, enabling more explainable and reliable decision making. In such a hybrid architecture, BNNs can help determine when predictions are sufficiently confident to act on, and when the system should refer to symbolic reasoning for more reliable or constrained decision making. This selective interplay between probabilistic and symbolic reasoning holds significant promise for enhancing the robustness, interpretability, and trustworthiness of intelligent parking and broader transportation systems.

Additionally, exploring more advanced Bayesian methodologies, including hierarchical Bayesian models and hybrid deep learning architectures, could lead to further improvements in uncertainty estimation and predictive performance [6,28]. Beyond parking prediction, the proposed framework can be extended to other smart mobility applications, such as real-time traffic forecasting, ride-sharing demand prediction, and public transport optimisation [11,25]. These extensions would further validate the applicability of uncertainty-aware models in various urban mobility contexts.

Furthermore, deploying BNN models on edge devices and within federated learning frameworks presents an exciting direction for enhancing scalability while preserving data privacy [4,16]. This approach could allow for decentralised, real-time learning and prediction in large-scale smart city infrastructures. Another crucial aspect for future work is the real-world deployment of these models in live parking management systems, followed by extensive user studies to assess their practical impact and usability. Understanding how drivers interact with uncertainty-aware predictions and measuring the effectiveness of BNN-based guidance systems in reducing search time and congestion would provide valuable insights for further refinement.

This research contributes to the growing field of uncertainty-aware predictive modelling in smart cities. By demonstrating the effectiveness of BNNs in parking availability prediction, it provides strong evidence supporting their broader adoption in intelligent transportation systems. Future developments should focus on refining Bayesian methods and integrating them into large-scale urban mobility frameworks to enhance efficiency and reduce congestion. The continued advancement of these models, coupled with real-world deployment and evaluation, will be critical in shaping the next generation of smart city transportation solutions.

## Figures and Tables

**Figure 1 sensors-25-03463-f001:**
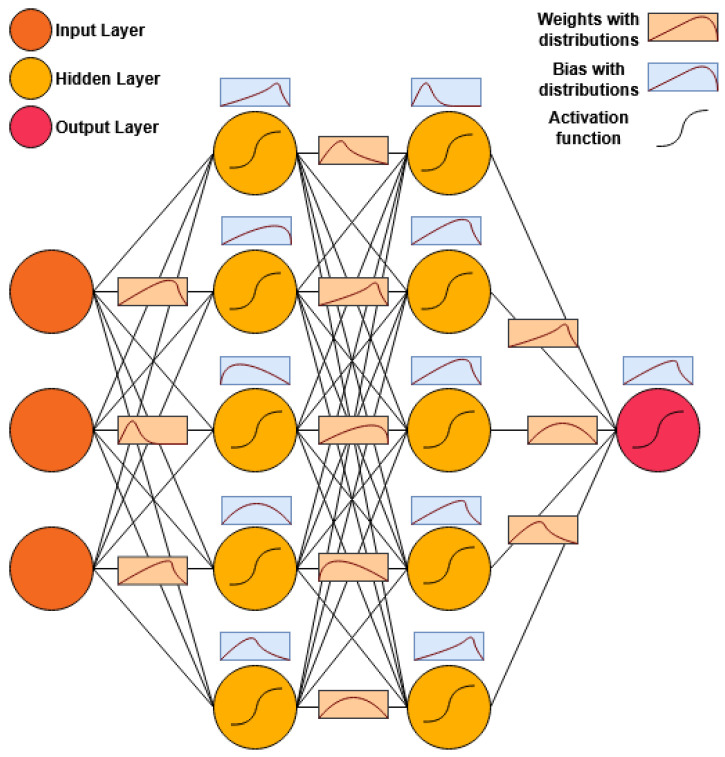
Architecture of a Bayesian Neural Network.

**Figure 2 sensors-25-03463-f002:**
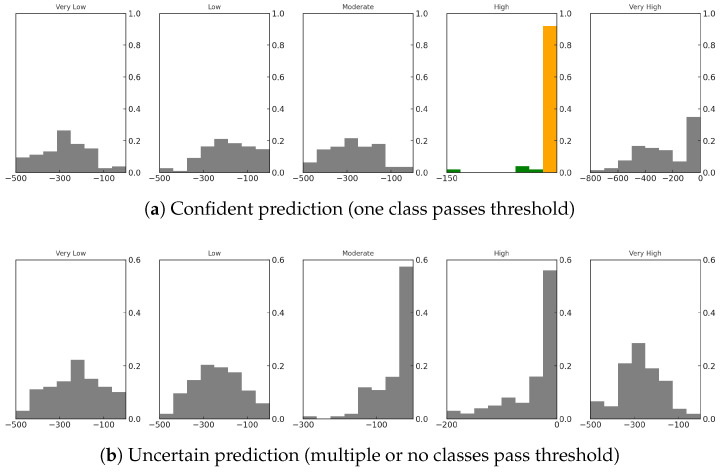
Examples of uncertainty distributions for a sample input. (**a**) A successful confident prediction where only one class passes the confidence threshold (The orange bar). (**b**) An uncertain case where no class exceeds the threshold or more than one class does.

**Figure 3 sensors-25-03463-f003:**
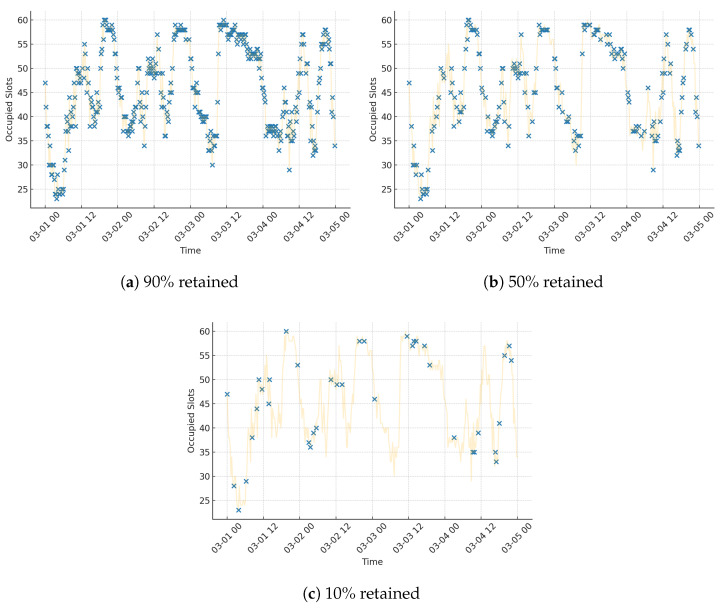
Visualising data scarcity for a single street segment (Franklin St between Queen St and Elizabeth St). Retained data points are shown in blue; the yellow faded line represents the full series. A total of (**a**) 90% data retained, (**b**) 50% retained, and (**c**) 10% retained.

**Figure 4 sensors-25-03463-f004:**
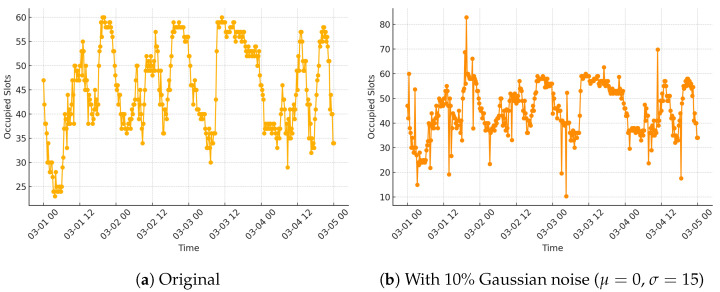
Simulating aleatoric uncertainty by injecting noise into occupancy data for a single segment. (**a**) Original occupancy over time. (**b**) Noisy version with 10% of values perturbed by Gaussian noise.

**Figure 5 sensors-25-03463-f005:**
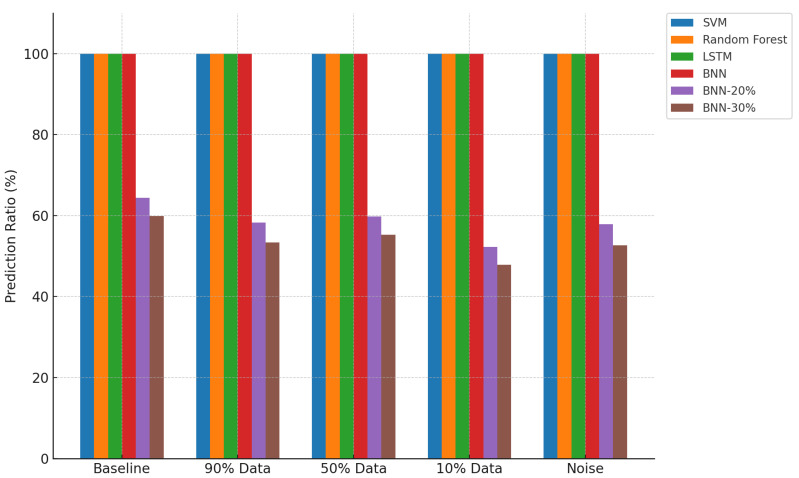
Comparison of prediction ratio across models and conditions.

**Figure 6 sensors-25-03463-f006:**
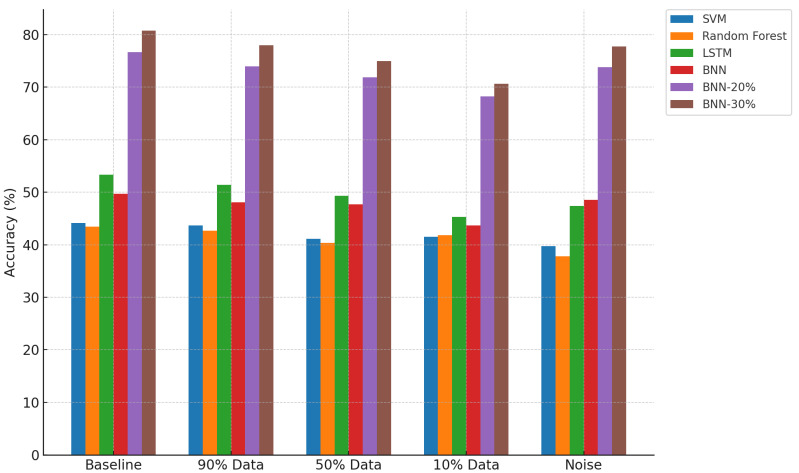
Comparison of accuracy across models and conditions.

**Figure 7 sensors-25-03463-f007:**
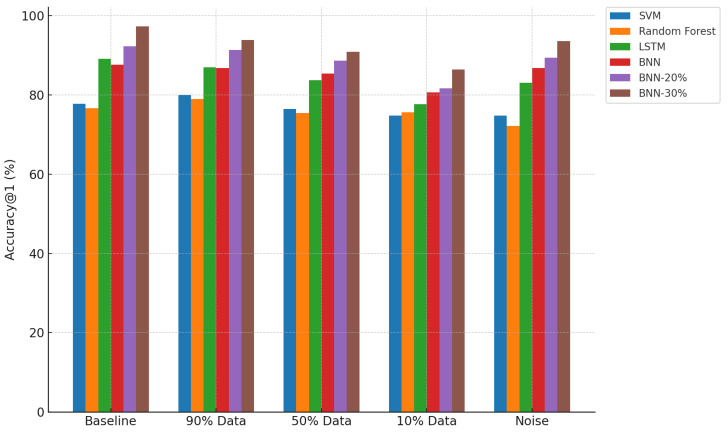
Comparison of accuracy@1 across models and conditions.

**Table 1 sensors-25-03463-t001:** Baseline Performance of SVM, Random Forest, LSTM, and BNN Models.

Model	Prediction Window 1			Prediction Window 2			Prediction Window 3		
	**Pred Ratio**	**Acc**	**Acc@1**	**Pred Ratio**	**Acc**	**Acc@1**	**Pred Ratio**	**Acc**	**Acc@1**
SVM	100.0%	46.2%	80.0%	100.0%	43.6%	76.9%	100.0%	42.7%	76.3%
Random Forest	100.0%	45.7%	79.3%	100.0%	43.3%	76.3%	100.0%	41.4%	74.5%
LSTM (Baseline)	100.0%	55.9%	90.2%	100.0%	52.3%	89.1%	100.0%	51.8%	88.0%
BNN (Baseline)	100.0%	51.7%	89.2%	100.0%	49.1%	87.5%	100.0%	48.2%	86.3%
BNN-20%	64.4%	78.3%	95.6%	60.2%	79.6%	90.5%	63.4%	72.0%	90.7%
BNN-30%	59.9%	83.3%	99.8%	58.6%	80.8%	95.7%	58.7%	78.1%	96.4%

**Table 2 sensors-25-03463-t002:** Performance under data scarcity.

Model	Prediction Window 1			Prediction Window 2			Prediction Window 3		
	**Pred Ratio**	**Acc**	**Acc@1**	**Pred Ratio**	**Acc**	**Acc@1**	**Pred Ratio**	**Acc**	**Acc@1**
**90% Data**									
SVM	100.0%	45.3%	82.2%	100.0%	42.7%	79.5%	100.0%	42.9%	78.3%
Random Forest	100.0%	44.8%	81.4%	100.0%	42.4%	78.9%	100.0%	40.9%	76.7%
LSTM	100.0%	53.8%	88.4%	100.0%	50.6%	86.9%	100.0%	49.8%	85.7%
BNN	100.0%	50.5%	88.3%	100.0%	47.7%	86.7%	100.0%	46.1%	85.4%
BNN-20%	58.3%	75.8%	92.9%	55.1%	77.3%	91.4%	57.1%	68.8%	89.8%
BNN-30%	53.4%	81.2%	96.8%	51.2%	79.1%	92.8%	53.6%	73.6%	91.9%
**50% Data**									
SVM	100.0%	43.3%	79.2%	100.0%	40.3%	75.9%	100.0%	39.8%	74.3%
Random Forest	100.0%	42.7%	78.4%	100.0%	39.9%	75.1%	100.0%	38.5%	72.7%
LSTM	100.0%	51.9%	84.8%	100.0%	48.7%	83.6%	100.0%	47.3%	82.7%
BNN	100.0%	49.7%	86.9%	100.0%	47.1%	85.2%	100.0%	46.3%	84.2%
BNN-20%	59.8%	73.6%	89.6%	57.9%	75.1%	88.5%	59.4%	66.9%	87.8%
BNN-30%	55.3%	77.4%	93.8%	53.4%	75.8%	89.9%	54.2%	71.7%	88.9%
**10% Data**									
SVM	100.0%	44.1%	77.1%	100.0%	40.8%	74.3%	100.0%	39.7%	73.1%
Random Forest	100.0%	44.7%	78.1%	100.0%	41.0%	75.1%	100.0%	39.7%	73.8%
LSTM	100.0%	48.3%	79.6%	100.0%	44.4%	77.3%	100.0%	43.2%	76.2%
BNN	100.0%	45.6%	82.7%	100.0%	43.7%	80.6%	100.0%	41.8%	78.8%
BNN-20%	52.3%	69.8%	84.3%	50.2%	71.1%	81.6%	50.9%	63.8%	79.2%
BNN-30%	47.8%	73.2%	88.4%	45.9%	71.4%	86.1%	46.8%	67.4%	84.7%

**Table 3 sensors-25-03463-t003:** Performance under noisy data.

Model	Prediction Window 1			Prediction Window 2			Prediction Window 3		
	**Pred Ratio**	**Acc**	**Acc@1**	**Pred Ratio**	**Acc**	**Acc@1**	**Pred Ratio**	**Acc**	**Acc@1**
SVM	100.0%	42.3%	78.0%	100.0%	39.4%	75.0%	100.0%	37.6%	71.5%
Random Forest	100.0%	40.0%	74.9%	100.0%	37.4%	71.7%	100.0%	36.0%	69.9%
LSTM (Noise)	100.0%	50.2%	84.6%	100.0%	46.7%	82.9%	100.0%	45.2%	81.6%
BNN (Noise)	100.0%	50.4%	88.2%	100.0%	48.1%	86.7%	100.0%	47.1%	85.4%
BNN-20% (Noise)	57.9%	75.8%	92.7%	55.1%	77.2%	88.5%	56.8%	68.5%	87.1%
BNN-30% (Noise)	52.7%	81.1%	96.6%	50.8%	78.9%	92.6%	52.9%	73.3%	91.6%

## Data Availability

Data is available in [31].

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
