# Peer review of "Uncertainty-Aware Parking Prediction Using Bayesian Neural Networks"

_sensors, 2025, doi:10.3390/s25113463_

Round 1

Reviewer 1 Report

Comments and Suggestions for Authors

This manuscript proposes an Uncertainty-Aware Parking Prediction approach using Bayesian Neural Networks. 

The topic is certainly relevant within the field of Intelligent Transportation Systems.

Also, addressing the uncertainty in the prediction process is equally significant in the broader context of enhancing the trustworthiness of Machine Learning models.

However, although the underlying idea is sound, the current version of the paper lacks several essential elements that, in my view, preclude its acceptance in its present form.

The primary concern pertains to how prediction uncertainty is handled. The authors mention having considered scenarios with data scarcity and injected noise to simulate epistemic and aleatoric uncertainties, respectively. While the general methodology is valid, the paper lacks a clear explanation of how these uncertainty components are assessed at the level of individual predictions.

Specifically, once the model is trained, one would expect each test prediction to be associated with a distribution of possible values, which can be attributed to either epistemic or aleatoric effects. In other words, for each prediction of the occupancy ratio, I expected to find a characterization in terms of both the mean value (as the best estimate) and the uncertainty (as the dispersion) of the predicted outcomes.

This level of analysis (uncertainty characterization per individual prediction) is not visible in the current version. Therefore, the uncertainty is not truly addressed in a detailed and rigorous manner.

Tables 2 and 3 report how the accuracy changes with data scarcity and noise; however, these metrics are presented as single scalar values, without any associated uncertainty. This is inconsistent with the core premise of the paper.

In conclusion, I strongly recommend the addition of a section that explicitly describes, after model training:

1. how each prediction is associated with an uncertainty (i.e., distribution of outcomes);

2. how predictions with uncertainty are then categorized into classes;

3. and how classification accuracy is assessed in the presence of the quantified uncertainty, separating and discussing epistemic and aleatoric contributions.

In addition, I provide a few further observations, which, although of minor importance, deserve attention:

While the manuscript claims to implement a Bayesian Neural Network, at line 275, the authors refer to the use of Monte Carlo Dropout. Therefore, it should be clarified whether a true Bayesian Neural Network has been implemented or whether an approximation via Monte Carlo Dropout has been used. This distinction is essential for methodological transparency.

Figure 1 does not seem to add significant value, as it is not clearly explained in the text. I suggest either improving its description or removing it.

In Figure 2, it would be advisable to clarify that each neuron includes a bias term, which, like the weights, is also a probability distribution. Additionally, please indicate the presence of the activation function in the neuron schematic.

Author Response

Dear Reviewer,

We thank you for your thorough evaluation and feedback.  

Each suggestion and comment raised by you has been carefully considered and addressed. Please find our responses to your comments and suggestions for our article titled "Uncertainty-Aware Parking Prediction Using Bayesian Neural Networks" below. (Comments appear in the left column of the table below; our responses are presented in the right column with specific references to sections in the revised manuscript).

Please note that, for clarity, we have prepared two versions of the revised manuscript: one with changes marked in blue for edits and red for new text, and another without colour markings.

Yours sincerely,

ALIREZA NEZHADETTEHAD, ARKADY ZASLAVSKY, ABDUR RAKIB and SENG W. LOKE

No

Comments/Suggestions

Response

1

While the general methodology is valid, the paper lacks a clear explanation of how these uncertainty components are assessed at the level of individual predictions.

Specifically, once the model is trained, one would expect each test prediction to be associated with a distribution of possible values, which can be attributed to either epistemic or aleatoric effects. In other words, for each prediction of the occupancy ratio, I expected to find a characterization in terms of both the mean value (as the best estimate) and the uncertainty (as the dispersion) of the predicted outcomes. This level of analysis (uncertainty characterization per individual prediction) is not visible in the current version. Therefore, the uncertainty is not truly addressed in a detailed and rigorous manner.

Tables 2 and 3 report how the accuracy changes with data scarcity and noise; however, these metrics are presented as single scalar values, without any associated uncertainty. This is inconsistent with the core premise of the paper. In conclusion, I strongly recommend the addition of a section that explicitly describes, after model training:

1. how each prediction is associated with an uncertainty (i.e., distribution of outcomes);

2. how predictions with uncertainty are then categorized into classes;

3. and how classification accuracy is assessed in the presence of the quantified uncertainty, separating and discussing epistemic and aleatoric contributions.

We sincerely thank the reviewer for this thoughtful and detailed comment, which helped us substantially improve the clarity and rigour of our explanation regarding uncertainty quantification at the inference stage.

To address your concerns, we have significantly revised Section 3.3 to provide a clearer and more comprehensive description of how uncertainty is assessed post-training, particularly at the level of individual test predictions. We now explicitly explain how each input undergoes 100 stochastic forward passes through the Bayesian neural network, each pass sampling from the variational posterior to reflect epistemic uncertainty.

From these samples, we generate 100 log-softmax outputs per input, which are exponentiated to obtain probability distributions across the five occupancy classes. For each class, we then compute the median (50th percentile) of the 100 sampled probabilities. This provides a robust central estimate of the model’s belief for that class, while also implicitly capturing the spread of uncertainty around it.

We further explain how this process informs the model’s predictive decision:

  • If a class's median probability exceeds a predefined confidence threshold (e.g., 0.2 or 0.3), it is considered as a valid output.
  • If only one class meets this criterion, it is selected as the prediction.
  • If multiple classes cross the threshold, or none do, the model withholds its prediction to reflect uncertainty—either due to class ambiguity or lack of confidence.

This decision-making logic directly supports uncertainty-aware classification and aligns with the principles of selective prediction and abstention in high-stakes or ambiguous scenarios. It also differentiates between cases of high confidence (low epistemic uncertainty) and cases of uncertainty, without forcing a hard prediction in every instance.

To further clarify this process, we have added a new figure (Figure 2) comprising two sets of visualised histograms across the five output classes for representative inputs. These histograms illustrate the dispersion of the model’s predictions for each class:

  • In Figure 2-a, the histograms show one class with a dominant and concentrated logit distribution, leading to a confident prediction.
  • In Figure 2-b, the distributions show greater overlap across multiple classes or no clear peak, demonstrating the type of scenarios where the model abstains from prediction due to high epistemic uncertainty.

These visual aids complement the textual explanation and provide an intuitive understanding of how uncertainty is characterised and used during inference. They directly address your request to clarify the treatment of uncertainty at the individual prediction level and to discuss how classification is performed in the presence of uncertainty.

We hope these improvements fully meet the expectations expressed in your comment, and we greatly appreciate your role in helping us strengthen this core part of the paper.

2

While the manuscript claims to implement a Bayesian Neural Network, at line 275, the authors refer to the use of Monte Carlo Dropout. Therefore, it should be clarified whether a true Bayesian Neural Network has been implemented or whether an approximation via Monte Carlo Dropout has been used. This distinction is essential for methodological transparency.

We thank the reviewer for highlighting this important point regarding methodological clarity. In the revised manuscript, we have clarified that the Bayesian Neural Network was implemented using Bayesian variational inference, not Monte Carlo Dropout. The paragraph in question (previously around line 275) has been rewritten to explicitly state the inference method used for approximating the posterior distribution over weights.

3

Figure 1 does not seem to add significant value, as it is not clearly explained in the text. I suggest either improving its description or removing it.

Upon review, we agree that Figure 1 did not add substantial value beyond the accompanying text and may have introduced redundancy. As recommended, we have removed Figure 1 from the revised manuscript to improve focus and clarity.

4

In Figure 2, it would be advisable to clarify that each neuron includes a bias term, which, like the weights, is also a probability distribution. Additionally, please indicate the presence of the activation function in the neuron schematic.

We appreciate the reviewer’s attention to detail. In the revised manuscript, we have updated Figure 2 to explicitly include the bias term as a probability distribution alongside the weights. We have also added the activation function to the schematic to fully represent the operation of a Bayesian neuron.

Reviewer 2 Report

Comments and Suggestions for Authors

This manuscript presents a Bayesian Neural Network for parking availability prediction. The authors compare their BNN approach with traditional LSTM networks under varying data conditions and demonstrate significant performance improvements.

  1. I suggest adding a quantitative comparison of training and inference times to provide readers with a more comprehensive understanding of the practical trade-offs involved.

  2. A more systematic analysis of how different threshold values affect the prediction ratio and accuracy would enhance the paper's contribution, i.e., why did you choose 20% and 30%?

  3. I suggest the authors compare the performance with other methods, not only the LSTM.

  4. Some references are dated. Please include more recent work on uncertainty quantification in transportation applications.

  5. The introduction could be enhanced by adding the specific research questions addressed in the paper.

Author Response

Dear Reviewer,

We thank you for your thorough evaluation and feedback.  

Each suggestion and comment raised by you has been carefully considered and addressed. Please find our responses to your comments and suggestions for our article titled "Uncertainty-Aware Parking Prediction Using Bayesian Neural Networks" below. (Comments appear in the left column of the table below; our responses are presented in the right column with specific references to sections in the revised manuscript).

Please note that, for clarity, we have prepared two versions of the revised manuscript: one with changes marked in blue for edits and red for new text, and another without colour markings.

Yours sincerely,

ALIREZA NEZHADETTEHAD, ARKADY ZASLAVSKY, ABDUR RAKIB and SENG W. LOKE

No

Comments/Suggestions

Response

1

I suggest adding a quantitative comparison of training and inference times to provide readers with a more comprehensive understanding of the practical trade-offs involved.

We thank the reviewer for this suggestion. While we acknowledge the importance of computational performance, our current focus is on evaluating predictive accuracy and the ability to quantify uncertainty under varying data conditions. We will consider incorporating training and inference time comparisons in future work or extended versions of this study where system-level efficiency is a primary concern.

2

A more systematic analysis of how different threshold values affect the prediction ratio and accuracy would enhance the paper's contribution, i.e., why did you choose 20% and 30%?

In the revised manuscript, we have clarified the selective prediction strategy and the rationale behind the chosen thresholds in Section 3.3. For each test instance, we perform 100 stochastic forward passes through the Bayesian neural network to obtain class probability distributions. We then compute the median probability for each class, and a prediction is made only if the highest median probability exceeds a predefined confidence threshold. We experimented with multiple threshold values and found that 20% and 30% provided the most effective trade-off between prediction ratio and accuracy. These updates provide a clearer justification for our threshold choices and highlight how confidence-based filtering improves reliability in uncertain conditions.

3

I suggest the authors compare the performance with other methods, not only the LSTM.

In the revised manuscript, we have extended our comparative analysis to include two additional baseline models: Support Vector Machine (SVM) and Random Forest (RF). These models were chosen to represent diverse machine learning paradigms beyond deep learning and were evaluated under all experimental scenarios, including baseline, data scarcity, and noisy conditions.

4

Some references are dated. Please include more recent work on uncertainty quantification in transportation applications.

We thank the reviewer for this helpful suggestion. In the revised manuscript, we have updated the Related Work section to include recent publications from 2024 that address uncertainty quantification in transportation applications (References 26 and 27). These additions strengthen the currency and relevance of the cited literature and ensure the manuscript reflects the latest developments in the field.

[26] Wang, Y.; Ke, S.; An, C.; Lu, Z.; Xia, J. A Hybrid Framework Combining LSTM NN and BNN for Short-term Traffic Flow Prediction and Uncertainty Quantification. KSCE Journal of Civil Engineering 2024, 28, 363–374.

[27] Sengupta, A.; Mondal, S.; Das, A.; Guler, S.I. A Bayesian approach to quantifying uncertainties and improving generalizability in traffic prediction models. Transportation Research Part C: Emerging Technologies 2024, 162, 104585.

5

The introduction could be enhanced by adding the specific research questions addressed in the paper.

In the revised manuscript, we have added a dedicated paragraph at the end of the Introduction section that explicitly states the research questions guiding this study. These questions clarify the scope, objectives, and contributions of our work on uncertainty-aware parking prediction using Bayesian Neural Networks.

Reviewer 3 Report

Comments and Suggestions for Authors

The authors propose the application of Bayesian neural networks to predict parking occupancy problems. Although the author points out that the proposed method has the advantage of considering model uncertainty, the experimental design did not fully demonstrate the superiority of the proposed method. Besides, there is not enough that just only LSTM be compared. The authors have to compare more algorithms in performance comparison.

The following issues should be modified:

  1. The authors point out that Ref. [15] is related to BNN research in line 56. However, [15] is not related to the application of BNN, and the authors need to clarify.
  2. The reviewer does not see how the authors process the uncertainty of the open-source dataset. The author can refer to Ref [2] method: Use visual data graphs to describe the uncertainty, noise, and data scarcity of the model in sections 3.1 and 3.4, respectively.
  3. The author needs to provide a detailed explanation of the prediction ratio index in the 3.5 section. The reason why the proposed method has good results seems to be due to the use of the prediction ratio, which allows the author to exclude samples that are difficult to classify, resulting in better accuracy indicators.
  4. The BNN model only outputs occupation levels for 5 categories. The author needs to clarify whether the occupation levels of the five categories are sufficient in practical applications.
  5. Please pay attention to the typesetting. Many paragraphs have a phenomenon that only one word in the last line.

Author Response

Dear Reviewer,

We thank you for your thorough evaluation and feedback.  

Each suggestion and comment raised by you has been carefully considered and addressed. Please find our responses to your comments and suggestions for our article titled "Uncertainty-Aware Parking Prediction Using Bayesian Neural Networks" below. (Comments appear in the left column of the table below; our responses are presented in the right column with specific references to sections in the revised manuscript).

Please note that, for clarity, we have prepared two versions of the revised manuscript: one with changes marked in blue for edits and red for new text, and another without colour markings.

Yours sincerely,

ALIREZA NEZHADETTEHAD, ARKADY ZASLAVSKY, ABDUR RAKIB and SENG W. LOKE

No

Comments/Suggestions

Response

1

 Although the author points out that the proposed method has the advantage of considering model uncertainty, the experimental design did not fully demonstrate the superiority of the proposed method. Besides, there is not enough that just only LSTM be compared. The authors have to compare more algorithms in performance comparison.

We thank the reviewer for this suggestion. In response, we have incorporated two additional baseline methods—Support Vector Machine (SVM) and Random Forest (RF)—to strengthen the comparative analysis. These models were selected for their widespread use in classification tasks and were evaluated under all experimental setups. The inclusion of these models offers a clearer perspective on how the proposed Bayesian Neural Network (BNN) framework performs relative to both deterministic classifiers and traditional deep learning baselines.

2

The authors point out that Ref. [15] is related to BNN research in line 56. However, [15] is not related to the application of BNN, and the authors need to clarify.

The issue stemmed from an incorrect citation key in the manuscript, which has now been corrected. Ref. [15] in the revised version of the paper now correctly points to a relevant study on Bayesian Neural Networks in transportation applications. The sentence in line 56 remains unchanged in content but is now properly supported by the correct reference.

3

The reviewer does not see how the authors process the uncertainty of the open-source dataset. The author can refer to Ref [2] method: Use visual data graphs to describe the uncertainty, noise, and data scarcity of the model in sections 3.1 and 3.4, respectively.

To address this comment, we have revised Section 3.4 to include visual illustrations of both data scarcity and synthetic noise injection. Specifically, we now provide subfigures showing the effect of reducing data availability (90%, 50%, and 10% retained) for a representative street segment, alongside another set of subfigures comparing the original occupancy signal with its noisy counterpart after Gaussian noise injection.

4

The author needs to provide a detailed explanation of the prediction ratio index in the 3.5 section. The reason why the proposed method has good results seems to be due to the use of the prediction ratio, which allows the author to exclude samples that are difficult to classify, resulting in better accuracy indicators.

We agree that the prediction ratio plays a central role in understanding the performance of thresholded BNN models. In the revised manuscript, we have substantially expanded Section 3.5 to provide a more detailed explanation of the prediction ratio as a coverage metric. We clarify that thresholded BNNs do not always produce predictions, but instead abstain from making decisions on uncertain inputs. This selective abstention is a deliberate and beneficial feature of uncertainty-aware modelling.

Rather than interpreting the improved accuracy of thresholded BNNs as an artefact, we now emphasise that it reflects the model's capacity to recognise and defer low-confidence predictions. These uncertain instances can then be passed to alternative components—such as symbolic reasoning systems or rule-based modules—thereby supporting more robust and adaptive decision-making. This clarification is now explicitly addressed in Section 3.5 of the revised manuscript.

5

The BNN model only outputs occupation levels for 5 categories. The author needs to clarify whether the occupation levels of the five categories are sufficient in practical applications.

The choice to discretise occupancy into five categories—Very Low (0–20%), Low (20–40%), Moderate (40–60%), High (60–80%), and Very High (80–100%)—is grounded in prior work and practical considerations. Specifically, Richter et al. [30] employed a three-category system (Low, Medium, High) based on percentage ranges, in line with the methodology used in the real-world SFPark project in San Francisco. Their justification was that relative availability classes are more useful and interpretable to users than absolute space counts, especially when users are unaware of the total capacity per road segment.

Building on this rationale, we extended the categorisation to five classes to provide finer granularity while preserving interpretability. This level of resolution helps distinguish between more nuanced occupancy levels (e.g., low vs. very low), which is useful for systems providing real-time guidance. It also enhances the expressiveness of our model’s predictive distributions without overwhelming decision-making logic or the end user.

To clarify this design choice, we have updated the last paragraph of Section 3.1 in the revised manuscript to reflect this reasoning and added a reference to Richter et al. [30].

6

Please pay attention to the typesetting. Many paragraphs have a phenomenon that only one word in the last line.

In the revised manuscript, we have carefully reviewed and edited all affected paragraphs to eliminate instances where a single word appeared on the final line. This was achieved by rephrasing sentences and adjusting line length to ensure consistent typesetting throughout the manuscript.

Reviewer 4 Report

Comments and Suggestions for Authors

This study proposed a BNN-based framework for parking availability prediction that explicitly quantifies both epistemic and aleatoric uncertainty. This is a meaningful study. However, the paper has a significant flaw: the authors compare their proposed BNN algorithm with LSTM, but the accuracy of predictions depends not only on the model type but also heavily on the model parameter settings. Different parameter settings for the same model can lead to significantly different prediction accuracies. The authors do not specify the model parameter settings used in their experiments, which makes the comparison less meaningful. The authors are advised to revise and supplement this information.

Author Response

Dear Reviewer,

We thank you for your thorough evaluation and feedback.  

Each suggestion and comment raised by you has been carefully considered and addressed. Please find our responses to your comments and suggestions for our article titled "Uncertainty-Aware Parking Prediction Using Bayesian Neural Networks" below. (Comments appear in the left column of the table below; our responses are presented in the right column with specific references to sections in the revised manuscript).

Please note that, for clarity, we have prepared two versions of the revised manuscript: one with changes marked in blue for edits and red for new text, and another without colour markings.

Yours sincerely,

ALIREZA NEZHADETTEHAD, ARKADY ZASLAVSKY, ABDUR RAKIB and SENG W. LOKE

No

Comments/Suggestions

Response

1

The authors compare their proposed BNN algorithm with LSTM, but the accuracy of predictions depends not only on the model type but also heavily on the model parameter settings. Different parameter settings for the same model can lead to significantly different prediction accuracies. The authors do not specify the model parameter settings used in their experiments, which makes the comparison less meaningful. The authors are advised to revise and supplement this information.

We thank the reviewer for this suggestion. In the revised manuscript, we have added a detailed description of the model architecture and hyperparameters to the second paragraph of Section 4.1 (Experimental Setup). This includes the number of layers, neurons, activation functions, optimiser settings, and training configuration. To ensure a meaningful and fair comparison, we first optimised the LSTM model through grid search and then applied the same architecture and hyperparameter settings to the Bayesian Neural Network wherever applicable. This alignment allows us to attribute performance differences to the probabilistic nature of the BNN rather than differences in model configuration.

Round 2

Reviewer 1 Report

Comments and Suggestions for Authors

The authors have adequately addressed all of my concerns in the revised manuscript.

Reviewer 3 Report

Comments and Suggestions for Authors

No further comment.